**Primer**

# Ten principles for reliable, efficient, and adaptable coding in psychology and cognitive neuroscience

Johannes Roth [1,2,6] ✉, Yunyan Duan [3,6], Florian P. Mahner[1,4], Philipp Kaniuth[1,2],
Thomas S. A. Wallis [3,5,7] & Martin N. Hebart[1,2,5,7]

Writing code is becoming essential for psychology and neuroscience research, supporting increasingly advanced experimental designs, processing of ever-larger datasets and easy reproduction of scientific results. Despite its critical role, coding remains challenging for many researchers, as it is typically not part of formal academic training. We present a range of practices tailored to different levels of programming experience, from beginners to advanced users. Our ten principles help researchers streamline and automate their projects, reduce human error, and improve the quality and reusability of their code. For principal investigators, we highlight the benefits of fostering a collaborative environment that values code sharing. Maintaining basic standards for code quality, reusability, and shareability is critical for increasing the trustworthiness and reliability of research in experimental psychology and cognitive neuroscience.

Navigating the modern research environment in experimental psychology and cognitive neuroscience can be challenging without programming experience. From experimental design, stimulus presentation, and data collection to data analysis, visualization, and publication—today, nearly every step involves working with software. Consequently, scientists with programming skills often have a distinct advantage: they can streamline their research processes, more easily handle complex data sets, and ultimately increase their research productivity.

The Open Science movement has amplified the importance of competent programming, by advocating for transparent, open, and reproducible research. In recent years, there has been a growing emphasis on sharing code and data to ensure the reproducibility of scientific findings and the reusability of scientific data[1–4]. However, many researchers prioritize the immediate outcomes of their code over its maintainability[5], which may stem from the pressures of the academic system and the challenges associated with producing and maintaining reproducible code. This may lead to code that is difficult to understand, reuse, or extend, creating significant barriers to reproducing and validating research findings and resulting in hidden costs and inefficiencies.

This paper aims to empower psychologists, cognitive neuroscientists, and researchers in related fields by introducing a set of simple yet effective programming practices inspired by professional software engineering. These principles are tailored to the typical workflow of a research group, from initial experimental design through data collection and analysis to publication. First, they lead to an efficient, shareable and reproducible approach with minimal repetitive work. Second, even for beginners, they can lead to high-quality research code that can benefit the broader scientific community by evolving into reusable research software. Finally, good programming practices encourage team collaboration and improve overall efficiency and effectiveness. Where applicable, code examples are given for both Python and R[6], two of the most commonly used (and non-proprietary) programming languages in psychology and neuroscience. While reproducible data and standards for data management are also important, here we focus on best coding practices but provide references for improving data reproducibility.

## Experimentation

### "Prototyping mode" vs. "development mode"

In psychology and cognitive neuroscience research, programming is typically carried out in a rather exploratory manner to solve specific problems. This approach fosters a habit of quick-and-dirty coding to achieve immediate, short-term objectives, such as plotting data or restructuring a

[1]Vision and Computational Cognition Group, Max Planck Institute for Human Cognitive and Brain Sciences, Leipzig, Germany. [2]Computational Cognitive Neuroscience and Quantitative Psychiatry, Department of Medicine, Justus Liebig University Giessen, Giessen, Germany. [3]Centre for Cognitive Science, Institute for Psychology, Technical University Darmstadt, Darmstadt, Germany. [4]Neural Coding Lab, Donders Institute for Brain Cognition and Behavior, Nijmegen, The Netherlands. [5]Center for Mind, Brain and Behavior (CMBB), Universities of Marburg, Giessen, and Darmstadt, Marburg, Germany. [6]These authors contributed equally: Johannes Roth, Yunyan Duan. [7]These authors jointly supervised this work: Thomas S. A. Wallis, Martin N. Hebart. ✉e-mail: jroth@cbs.mpg.de

table. We refer to this as "prototyping mode", which is characterized by a focus on speed and experimentation. In this mode, once the code appears to work as expected, the researcher moves on to the next task. Prototyping mode makes sense in many situations: it often does not pay off to put effort into writing high-quality code that will never be used again, especially when the analysis is exploratory and might not yield the desired outcome. In these cases, prototyping mode is a practical and efficient approach.

However, relying too heavily on prototyping as a default mode can easily lead to significant problems down the line: Code that contains errors can invalidate scientific conclusions, poorly written code and lack of documentation can complicate collaboration, and unstructured codebases can make replicating results unnecessarily difficult. For these reasons, we urge researchers to regularly switch to "development mode". In this mode, following the rapid initial prototyping phase, code is brought to a higher standard, ensuring correctness, modularity, reusability, and shareability. Ideally, this transition should occur not just before publication, where the cost of correcting mistakes may be too high, but throughout the duration of a project, with alternating phases of prototyping and development.

Inspired by related general guidelines for researchers[7–12], we examine which practices psychologists and cognitive neuroscientists should consider as they switch to "development mode". In the following sections, we present real-world scenarios commonly encountered by psychologists operating in "prototyping mode" and offer practical recommendations for addressing the challenges that arise from this approach.

Box 1 briefly summarizes our ten principles of coding practices. To illustrate how these could be applied across operating modes, Fig. 1 provides an overview of a few possible stages a research psychologist might go through during their study, along with corresponding coding practices.

### Organizing code for automation and reproducibility

The research process often consists of repetitive tasks or procedures that depend on specific knowledge or resources. For example, for preprocessing of neuroimaging data, it is necessary to perform a number of steps in a certain order, which is often done by manually clicking a number of buttons in a graphical user interface. Likewise, in behavioral studies, results from individual participants are often manually combined into a larger table. These tasks are highly repetitive and error-prone. Automating such tasks can quickly be learned (even by beginners), significantly reducing the risk of human error and saving a lot of time. In addition, replacing manual elements with well-organized code can significantly streamline the workflow, making it easier to reuse and share code and thus improving the reproducibility of analyses.

Best practices for organizing code to support researchers in their ability to automate routine tasks and increase their efficiency are presented in Principles 1, 2, and 3.

### Writing reusable research code

Although research code only rarely has to meet the rigorous standards of professional software development, adhering to some basic principles of software development can help minimize potential issues: These principles ensure that code is easily understood, especially by the author's future self, functions as intended, and is easily usable by others.

Principles 4, 5, and 6 introduce practices aimed at improving code quality.

### Programming in a collaborative setting

Research is often a team effort, with individual projects contributing to the larger agenda of a research group. In this context, good coding practices are particularly beneficial and facilitate collaboration and teamwork among lab members. However, collaborative coding demands a coordinated effort by the whole team, from students to Principal Investigators (PIs).

Perhaps most importantly, PIs should be aware that they have a significant influence on the motivation and capacity of their team to incorporate these practices into their daily work. They should articulate a clear vision of programming's role in the group and actively support their

---

## Box 1 | Ten principles for research programming

These are the ten main programming principles discussed in this paper.

**Organizing code**
Principle 1: Adopt sensible standards.
Principle 2: Track changes.
Principle 3: Accelerate the research workflow.
**Writing reusable research code**
Principle 4: Write "good" code.
Principle 5: Test your code.
Principle 6: Think about others.
**Programming in a collaborative setting**
Principle 7: Seek awareness and consensus.
Principle 8: Conduct code reviews.
Principle 9: Build a shared knowledge base.
Principle 10: Consider infrastructure.

---

members accordingly. In addition, lab members assuming leadership roles in collaborative coding should carefully craft team-level programming policies. Last but not least, students are encouraged to actively approach their PIs about these practices and discuss if implementing them makes sense in their lab. This proactive approach can lead to improved coding practices and more efficient collaboration within the research group.

Principles 7, 8, 9, and 10 present best practices for programming collaboratively.

### Application
#### Principle 1: Adopt sensible standards
*Scenario: A senior colleague shares their code, advising you to adjust it to your project. You find a mess of disorganized scripts and undocumented data files, but you power through and get the analysis to run—only for the results to not make any sense. After many more hours of debugging, you finally find a typo in a custom function your colleague wrote that subtly alters the computation. When you inform your colleague, their eyes light up in horror, as they now need to double-check all results from their last paper.*

Code that is not clearly structured or that does not highlight dependencies and other code requirements can hinder its reusability. Reluctance to use established functions or software packages for basic functionality compounds the issue. The following practices can help prevent such scenarios.

**Use a standardized directory structure**. Adopting a standardized directory structure from the outset of a project is highly beneficial for organizing data and code. This consistency allows researchers to quickly locate and access the resources they need and facilitates clarity among team members and collaborators. A directory structure can be based on an existing template and then adjusted to best suit the specific needs of an individual or a research group. Some research fields have established standards, such as BIDS for neuroimaging data (https://bids.neuroimaging.io/index.html)[13], and incorporating these into the workflow is recommended. Standardizing the directory structure of studies within a research group promotes consistency across projects and allows group members to easily understand the structure of their colleagues' work (Fig. 2).

Beyond the immediate benefits of simplifying the workflow for coding, a standardized directory structure also benefits the management of research data. While an extensive discussion of research data management is beyond the scope of this guide, for comprehensive guidance we direct readers to established guidelines, such as the FAIR principles[14], as well as actionable best practices[15–17].

**Fig. 1 | Research stages and relevant operating modes and programming practices.** Depending on the current progress within a research project, different practices might be more important than others. Generally, the development mode should cycle between prototyping and development as a project progresses.

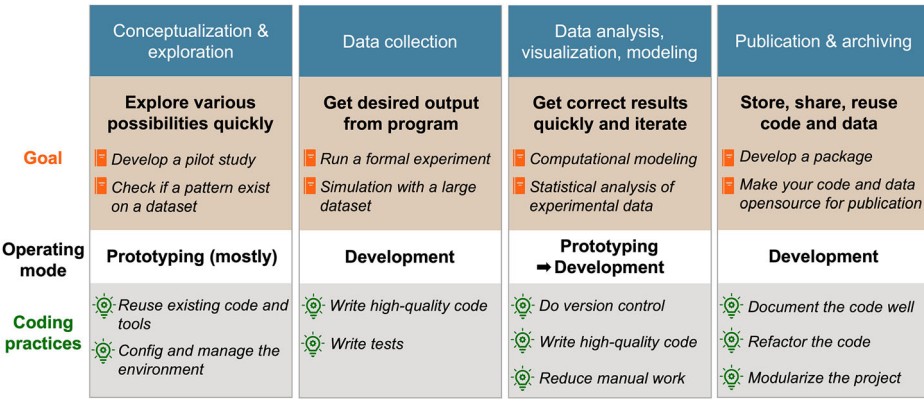

```
./{{cookiecutter.repo_name}}
├── README.md              <- Provide overview of the project with setup instructions and usage details.
├── data                   <- Store data from experiments, input data for models, and stimuli.
│   ├── lab_book.md        <- Keep a chronological lab notebook documenting data collection.
│   ├── processed_data     <- Store processed data after costly data pre-processing steps.
│   ├── raw_data           <- Store raw, unprocessed data from experiments or other sources.
│   └── stimuli            <- Save stimuli files for the project that are not generated on the fly.
├── docs                   <- Provide documentation for the package or codebase (e.g. generated by Sphinx).
├── figures                <- Save automatically or manually generated figures for the project.
├── publications           <- Keep all materials related to publications or reports.
├── results                <- Save outputs such as trained models, analysis output, checkpoints, pickle and hdf5 files.
├── scripts                <- Put standalone scripts (e.g. Python, R, Bash) and notebooks (e.g. Jupyter, Quarto) here.
├── src                    <- Save source code of your project and reuse the code as a package if possible.
└── tests                  <- Write tests to ensure the correctness of the code.
```

**Fig. 2 | An exemplary repository structure that could be used by a lab conducting experimental psychology research.** This directory structure includes a README file as an overview, a folder for tests, and separate subdirectories for data and scripts. Code that is used across many different aspects of the project (for example, for loading data) is located in its own directory (*src*). This is based on a general template for working with data (from https://github.com/drivendata/cookiecutter-data-

science). For research-specific frameworks, various examples can be found online (e.g., https://gin-tonic.netlify.app/standard/). Additional recommendations for sensible project-specific directory structures can be found in *The Good Research Code Handbook*[58].

In addition to adopting a standardized directory structure, paying attention to file naming conventions can pay off in the long run. Descriptive and comprehensive file names can be very useful, especially when files are moved outside of their original directory. For example, if a file named *behavioral.csv* in the directory *project_name/data/raw_data/sub-01* is accidentally misplaced, it will be difficult to impossible to relate it back to the participant or even the project itself. By contrast, a name like *project_name_sub-01_behavioral_raw.csv* conveys the file's content, subject, and origin.

**Configure and save the programming environment**. To ensure that code runs as intended and produces the same results every time it is repeated, it must be executed in a specific computing environment. This environment includes everything from the specific operating system to the packages and libraries used by the code, including their versions and all necessary configuration settings (see Box 2). Without an exact replica of this environment, reproducing results may be difficult, if not impossible. Therefore, keeping track of the environment is crucial for the reproducibility of scientific research.

There are many solutions to manage an environment. The simplest solution is creating a detailed list of all external packages used and their versions, but this would require a manual setup of the environment every time it is reused, which is error-prone and time-consuming. Each programming language offers specialized tools for dependency management, such as *conda* (https://github.com/conda-forge/miniforge)[18] for Python or *packrat*[19] for R. Alternatives to *conda*, such as *Mamba*[20], *Poetry*[21], and *uv*[22] are also becoming popular for faster package installation and/or better package resolution.

For a more comprehensive solution, so-called container tools like Docker[23] or Singularity[24] allow a fully encapsulated environment that even includes the operating system. This allows for the exact setup to be easily

shared and reproduced on any system. Since working with containers can be complex, we recommend starting with more basic environment management solutions. For neuroimaging software, we suggest Neurodesk as a container-based solution that works out of the box for most common neuroimaging software[25].

**Prefer existing tools and do not reinvent the wheel**. For many tasks researchers aim to accomplish, there are already existing tools and libraries. If a toolbox already offers a functionality required for analysis, it is generally not advisable to create a custom solution from scratch, unless the existing option does not fulfill the project's specific requirements, such as speed or compatibility with existing code. Consulting with colleagues who have faced similar issues can be invaluable, especially for more specialized or uncommon problems, since they may have already developed code that can be used or adapted.

Notable examples of existing tools include toolboxes designed for specific purposes, such as fMRI preprocessing (e.g., FSL[26], SPM[27]) or stimulus presentation (e.g., Psychtoolbox[28], PsychoPy[29]). Even for custom pipelines, a wealth of well-maintained open-source libraries are available. These libraries are usually hosted on public code repositories like GitHub, allowing examination of the code if issues arise or even contacting the creators for support.

For R users, established libraries are available on CRAN (Comprehensive R Archive Network; https://cran.r-project.org/), with well-known examples including *Tidyverse*[30] and *dplyr*[31] for data manipulation, *ggplot2*[32] for data visualization, and *caret*[33] for machine learning. Python users can find a wide array of packages on PyPI (Python Package Index; https://pypi.org/) and through volunteer-maintained *conda* channels, such as conda-forge (https://conda-forge.org)[18], both of which are repositories used by package managers. Conda channels maintained by volunteer communities are free

## Box 2 | Glossary

Gaining an understanding of basic programming terms and concepts is advisable. It helps streamline interactions with technical tools and improves communication with colleagues. This box outlines common programming terms used in this paper.

**Environment**—A general term to summarize the software and hardware components that code requires in order to run (the "dependencies"). Dependencies include but are not limited to the operating system, the names and versions of included third-party libraries, the programming language, and so on. Multiple environments can exist on the same computer, i.e., one script can use one version of a programming library while another script can use another version.

**Functions**—Functions group a set of commands to perform a specific task. Most functions take one or more inputs (called parameters or arguments), process these inputs, and return an output. Their internal processes are usually "hidden" from the script that users are using the functions in so that one can focus on inputs to outputs. Depending on the programming language, functions can have different levels of flexibility - some functions can even call themselves recursively.

**High-performance computing (HPC) cluster**—An HPC cluster is a collection of many separate yet interconnected computers typically used to solve computational problems that would take too long to run on a local computer.

**IDE**—The "Integrated Development Environment" (IDE) is the editor that people actually write code with. There are many different IDEs that fill different roles, but a good general-purpose and language-agnostic editor that is easily extensible is VSCode or VSCodium. One may want to choose a specialized IDE depending on the programming language of choice (RStudio for R, Matlab for Matlab).

**Notebook**—Notebooks are interactive tools to write and run code in small sections, where results are immediately displayed and can be combined with text and images. They are a great tool for quick experimentation and make it easy to combine code with its output in a single "literate" document. Popular tools include Jupyter Notebook, R Markdown, and Quarto.

**Scripts**—Scripts represent a series of commands (e.g., functions) within a program that are executed and interpreted one at a time. They are used in programming to automate repetitive tasks, manipulate data, and perform other complex functionalities.

**Source code**—Source code, also referred to as "code", is a collection of computer instructions, written in a human-readable programming language. This code gets compiled or interpreted in order to create an executable program. It is the base upon which all software is built and generally determines the behavior of the program.

**Shell/terminal/command line interface (CLI)**—The shell is a program that provides an interface with the operating system (e.g., Windows, Linux, Mac OS) and sends commands to it to execute. The terminal is a graphical interface for the shell. A command line interface (CLI) is a way of running programs using text commands, often through a shell. A shell/terminal/CLI allows one to efficiently interact with files and directories on one's PC, use command line tools, and much more. Basic knowledge of using a shell is a prerequisite for working with remote computers (e.g., compute clusters) and is also very handy for automating otherwise daunting tasks (e.g., renaming thousands of files).

**Absolute/relative file paths**—A file path references the location of a file within a directory structure. Absolute file paths specify the full location of the file, starting from the computer's root directory (e.g., */home/user/data/survey.csv* on a Linux-based system). On the other hand, relative file paths specify a file's location relative to the location where the file path is defined. For example, if a script located in */home/user/project/scripts/script1.py* specifies the relative file path *./data/survey.csv*, it will refer to a file located in */home/user/project/data/survey.csv*. If the whole project is moved to a different computer, the relative file path will still point to the correct file, whereas the absolute file path may no longer be correct as it depends on the structure of the new file system.

to use by anyone, whereas the default Conda channel (managed by Anaconda, a for-profit organization) has required payment from certain organizations since 2020[34]. Essential Python packages include *NumPy*[35] and *SciPy*[36] for numerical computations, *Pandas*[37] for data frame manipulation, *Matplotlib*[38] and *Seaborn*[39] for data visualization, and *scikit-learn*[40] as well as *Pytorch*[41]/*TensorFlow*[42] for machine learning and deep learning, respectively.

### Principle 2: Track changes

*Scenario: You are developing a computational model for predicting human decision-making and initially using a reinforcement learning model. Excited by a paper on a novel Bayesian approach, you decide to overhaul your codebase, overwriting and deleting old functions. The initial tests are promising, but weeks later you realize that the Bayesian approach struggles with certain scenarios the original model handled well. When you want to compare both approaches, you realize you have permanently overwritten the previous version, and there is no way to recover it.*

Research often involves exploring unknown outcomes, and therefore, frequent changes in code are very common. It can be tempting to just keep working and overwrite code in the process, specifically when one sees the change as an improvement. But this practice can be disastrous if reverting to an earlier state is necessary. To prevent this, we recommend creating a searchable and revertible history of code.

This can be easily achieved using a version control system, which automatically manages changes to files, provides access to previous versions of code, and allows files to be reverted to earlier states when needed. The most prominent example is Git (https://git-scm.com/about). A researcher who works alone can use just three simple commands to track changes in their day-to-day work:

```
$ // choose file(s) to be added to Git
$ git add method_xyz.py method_xyz.R
$ // commit and describe changes made
$ git commit -m "Implemented method XYZ"
$ // upload changes to a predefined server
$ git push
```

With 'git add' and 'git commit', Git keeps a record of all code changes in a so-called "repository", which acts as a virtual storage space for the project. With 'git push', changes can be saved to a remote repository. This remote repository acts as a central and up-to-date version of the project and can be hosted either on a self-managed server or online hosting services for Git repositories, like GitHub or GitLab. It allows collaboration among multiple researchers by "cloning" this repository to their local computer, making changes, and pushing their updates back to the central repository. In this way, code changes made by different people can be isolated, reviewed, and consolidated with ease. More details about the mechanisms of version control systems and how to best use them can be found in the section "Make incremental changes" in Wilson et al.[11].

Even if code history is preserved by a version control system, or as a simple alternative if Git seems too complicated, creating a separate "archive" folder for obsolete code can be useful. This way, instead of having to search through the repository's history, outdated code is immediately available when it is needed again.

## Box 3 | Best use of AI code assistants

Large language models (LLMs) like OpenAI's GPT, Anthropic's Claude, Meta's Llama, or Google's Gemini can generate text of many different kinds - including code. Not surprisingly, these AI tools have found widespread adoption among software developers, whether through IDE extensions like GitHub Copilot, TabNine, or Cursor, or chat interfaces like ChatGPT. These AI code assistants can be incredibly powerful, but we advise great caution if you choose to use them.

The models that power these tools are trained to predict the next token—a part of a word or a symbol—based on previous ones, essentially capturing the statistics of human-generated texts. With massive amounts of text and some advanced techniques, this simple rule of next token prediction has surprisingly been sufficient for generating fluent text, providing answers to questions, and even writing advanced code. Although these methods brought with them tremendous advances in their capability and usability, the basic mechanism behind the models makes their outputs also susceptible to error and confabulation, and these models will often double down on their incorrect answers. For coding, this means that, although the human-likeness and apparent accuracy and confidence of their output can give the illusion that there is logical reasoning involved in their generation[59], we need to remember that the mechanism is based on exploiting statistical patterns in text, and we cannot be sure that generated code is actually correct or logically sound. At the time of writing this guide, this holds even for models claimed to be capable of advanced reasoning.

Generated code must, therefore, always be carefully reviewed to ensure that it functions as intended.

Simply accepting it will sooner or later introduce bugs that can be hard to spot, especially if the errors are conceptual and do not result in crashes. This poses a problem for beginners, as it is especially tempting for them to let an AI write much of the code from scratch, even when they do not know what the code does or how it works. This means that if they lack the relevant experience to properly review and understand it, such errors can easily fall through the cracks. It is, therefore, critical to fully understand what the generated code does, to avoid blindly accepting it, and to use test-driven development to ensure it functions correctly. The impact of this on the ability of trainees to learn how to code is currently unknown,

as coding assistants may support their ability to improve their coding skills, but may also hamper their development of advanced understanding. In addition, even advanced users need to learn how to best prompt the AI tool to produce the desired result, which requires practice and experience.

Despite these issues, LLMs can be very useful for generating code in settings where you have a clear picture in mind of what you would like to accomplish and how to implement it. In this case, you can leave much of the actual implementation to the LLM and then test if the code and the results match your expectations—basically using the LLM as an intelligent autocomplete. In addition, LLMs can also be helpful for more open-ended problems, like:

- learning about frameworks or packages, (well-known) technical concepts, programming languages, or tools and letting it quiz you on your understanding
- explaining a seemingly undecipherable script that your coworker sent you, step by step
- giving tips on how to refactor your code and improve its performance
- converting a script written in Matlab to Python (given that you wrote tests to ensure that functionality stays intact)
- writing docstrings for functions or general documentation

As these tools further improve and become more capable and reliable, they will surely find increasing application in the sciences. Used carefully, they can already improve productivity, but it should also be noted that they have other issues besides confabulation. Privacy is a concern, as any input that is given to the model is sent to the servers the models are run on and, depending on the provider and user settings, can even be used in future model training. They can also cause copyright issues, as they readily reproduce publicly available code used in their training, regardless of its licensing. More broadly, an argument can be made that current LLMs are a form of "automated plagiarism"[60] or intellectual property theft, because they are trained on data that many creators did not give permission to be used in this way, and those creators cannot be credited. These topics should be discussed within a research group to delineate the group's policy on the responsible use of AI tools.

### Principle 3: Accelerate the research workflow

*Scenario: You are currently pre-processing MEG and EEG data from 30 participants, spending long hours in the lab manually entering it into a pre-processing tool. Afterward, you analyze all participants' data on your laptop, which takes about ten days. As you check the results, the first subject's results align with your pre-registered hypothesis, but the results do not make sense for the remaining subjects. After spending some time scrutinizing your analysis code, you reopen the preprocessing tool and realize that you mismatched file paths between MEG and EEG data for all but the first participant.*

When used effectively, programming serves as an incredible tool for automating tasks. Errors can easily creep into repetitive tasks when they are carried out manually. Whether it is relying on a slow laptop instead of a high-performance workstation or checking code line-by-line instead of using debugging tools, inefficiencies can result in prolonged wait times and needless work. This section elaborates on how automating manual steps, using the right resources, and leveraging proper tools can streamline the workflow, making it faster and more accurate.

**Automate manual steps**. Automating repetitive tasks not only saves time but also makes it easier to reproduce research findings. A good starting point is writing small scripts that handle analyses or pre-processing steps. For example, instead of manually entering file paths into a toolbox's graphical user interface (GUI), one can easily write a script to

iterate over all files and execute the corresponding GUI commands from the command line. This can be accomplished using the command line interface or a preferred programming language.

In more sophisticated scenarios, automation can be conditional, such as preprocessing data only if it has not been processed previously or if changes have occurred in the preprocessing itself. Language-agnostic tools like makefiles or, specifically tailored to scientific research, Snakemake (https://snakemake.github.io/)[43], serve as helpful resources for this purpose.

**Focus on efficiency and use the right resources**. Being mindful of how long different parts of the code take to run can mean the difference between waiting a week for results or only ten minutes. When an analysis takes a long time to run, it is worth inspecting its resource usage to identify processing bottlenecks, a process called profiling. Often, timing individual functions can be enough to pinpoint what needs optimization. In Python and R, this can be as simple as adding a few lines of code:

```python
import time
start_time = time.time()
register_brain_to_reference(brain, reference)
end_time = time.time()
print(f"Execution time: {end_time - start_time} sec
```

## Box 4 | "Bad" code vs. "good" code

Whether code is good or bad is often in the eye of the beholder. Still, there are principles of good code that are easy to spot. To illustrate what difference it can make to follow the advice from the sections on code quality, consider the following example written in Python and R:

```python
```python
1:  def f1(x):
2:      s = 0
3:      c = 0
4:      for i in x:
5:          s += i
6:          c += 1
7:
8:      return s/c
9:
10:  data = [1.2, 1.4, 1.3, 1.5,
1.2, 1.3, 1.4, 1.6, 1.3, 1.2]
11: print(f1(data))
```
```

```r
```R
1:  f1 <- function(x) {
2:    s <- 0
3:    c <- 0
4:    for (i in 1:length(x)) {
5:      s <- s + x[i]
6:      c <- c + 1
7:    }
8:    s / c
9:  }
10:  data <- c(1.2, 1.4, 1.3, 1.5,
1.2, 1.3, 1.4, 1.6, 1.3, 1.2)
11:  print(f1(data))
```
```

At first glance, it might take some time to figure out that this code simply computes the mean of the data in the array. It might also be surprising to learn that these values are actually response times collected in a behavioral experiment. By following best practices, multiple aspects of this small snippet can be improved.

- the calculation of the mean is overly complicated (lines 2-8)
- the function and variable names are not descriptive (lines 1, 2, 3, 10)
- there is no documentation explaining the purpose of the code

By addressing these issues, we might end up with the following code:

```python
```python
def
calculate_mean_reaction_time(reaction_time
s: List[float]) -> float:
    """
    Calculate the mean of a list of
reaction times.

    Args:
    reaction_times (list of float): The
reaction times to calculate the mean of.

    Returns:
    float: The mean of the reaction times.
    """

    total_reaction_time =
sum(reaction_times)
    number_of_reaction_time =
len(reaction_times)
    mean_reaction_time =
total_reaction_time /
number_of_reaction_time

    return mean_reaction_time

observed_reaction_times = [1.2, 1.4, 1.3,
1.5, 1.2, 1.3, 1.4, 1.6, 1.3, 1.2]

print("Average reaction time: ",
calculate_mean_reaction_time(observed_reac
tion_times))
```
```

```r
```R
calculate_mean_reaction_time <-
function(reaction_times) {
  #' Calculate the mean of a list of
reaction times.
  #' @param reaction_times: The reaction
times to calculate the mean of.
  #' @return The mean of the reaction
times.

  total_reaction_time <-
sum(reaction_times)
  number_of_reaction_time <-
length(reaction_times)
  mean_reaction_time <- total_reaction_time
/ number_of_reaction_time
}

observed_reaction_times <- c(1.2, 1.4, 1.3,
1.5, 1.2, 1.3, 1.4, 1.6, 1.3, 1.2)

print(paste("Average reaction time:",
calculate_mean_reaction_time(observed_react
ion_times)))
```
```

This function's purpose and usage are immediately clear: It uses explicit variable names and documentation, and it is easy to see that the calculation of the mean is correct. This example is only meant for illustrative purposes, though, as it does not make sense to reinvent the wheel - most basic math functions are already available in existing libraries—and it is overly verbose for a simple function.

Another example is a function that serves more than one purpose.

```python
def plot_accuracy(df):
    accuracy_bar_plot =
df.groupby("condition")\
    .agg(accuracy = ("is_correct", "mean"))\
    .reset_index()
    .plot.bar(x = "condition", y = "accuracy")

    return accuracy_bar_plot
```

```R
plot_accuracy <- function(df) {
  df |>
    group_by(condition) |>
    summarise(accuracy = mean(is_correct))
  |>
    ggplot(aes(x = condition, y = accuracy))
+
    geom_col()
}
```

This function first computes the average accuracy across conditions and then visualizes the results as a bar plot. Chaining these actions together may be convenient for a one-off analysis script, but for a function intended for reuse, errors may occur when computing the mean or creating the plot, leading to difficult debugging.

We can separate these steps into multiple functions to improve the clarity and maintainability of the function (comments and docstrings were omitted for brevity):

```python
def plot_accuracy(df):
    mean_accuracy_df = summarize_accuracy(df)
    accuracy_bar_plot =
plot_bar_mean(mean_accuracy_df)

    return accuracy_bar_plot

def summarize_accuracy(df):
    grouped_df = df.groupby("condition")\
    .agg(accuracy=("is_correct", "mean"))\
    .reset_index()
    return grouped_df

def plot_bar_mean(df):
    return
df.plot.bar(x="condition",y="accuracy")
```

```R
plot_accuracy <- function(df) {
  mean_accuracy_df <-
summarize_accuracy(df)
  plot_bar_mean(mean_accuracy_df)
}

summarize_accuracy <- function(df) {
  df |>
    group_by(condition) |>
    summarise(accuracy =
mean(is_correct))
}

plot_bar_mean <- function(df) {
  df |>
    ggplot(aes(x = condition, y =
accuracy)) +
    geom_col()
}
```

```
onds")
```
```R
start_time <- Sys.time()
register_brain_to_reference(brain, reference)
end_time <- Sys.time()
print(end_time - start_time).
```

Figuring out how to best optimize time-consuming code varies depending on the problem at hand, and there is no easy one-size-fits-all solution. Sometimes, rethinking an approach is necessary, like using vectorized operations over nested for-loops. In other cases, it is about using resources more effectively, as seen in instances where an easily parallelizable operation like motion correction on multiple fMRI files is programmed to run sequentially, or when researchers use their personal laptop instead of a computer cluster to run more time-intensive tasks. Another common case is when data goes through many processing stages and is processed from scratch every time the code is re-run, although intermediate processing steps could be easily saved and later loaded. For a detailed exploration of optimizing performance in Python and R, we recommend the books *High Performance Python: Practical Performant Programming for Humans*[44] and *Efficient R programming: A Practical Guide to Smarter Programming*[45], respectively.

Keeping an eye on efficiency is also advisable when testing code. It is not uncommon for a scientist to launch a long-running job before leaving the office, only to return the next day and find that the code crashed at the final step of saving the results. A smarter approach is to first test the code on a smaller data subset or run it for only a few iterations. The process can be even more efficient when the code is modularized and thus allows individual components to be tested separately.

**Make use of proper IDEs and debugging tools**
A wealth of tools exists for programming, but probably most essential to all developers is the Integrated Development Environment, or IDE (Box 2).

More than just a basic text editor, modern IDEs provide many useful capabilities that can be customized to specific needs. They provide helpful descriptions and tooltips of functions and packages, integrate tools for version control such as Git, and sometimes even offer code generation features (Box 3). We strongly recommend working with a proper IDE rather than relying solely on a text editor.

One of the most powerful features of IDEs is the debugger. A debugger allows stepping through code line-by-line while it is executed, pausing at a chosen line of code to inspect the values of all variables. Compared to scattered print statements throughout the code, using a debugger is much more effective at identifying bugs.

## Principle 4: Write "good" code
*Scenario: You quickly create Python code to collect reaction times for a study on social media's impact on cognitive functions. Upon analyzing the results, you notice that the recorded reaction times are suspiciously uniform—a list of integers in descending order. You realize you accidentally used the variable 'rt' for both reaction time and remaining trials, causing reaction times to be overwritten with a countdown after each trial. This forces you to discard all data and repeat the experiment, setting your research back by several weeks.*

Concrete advice on how to write "good" code is abundantly available, with valuable resources in books, blog articles, and tutorial videos. While the definition of "good" code varies across programming languages and styles, there are universal properties that every developer values[46], and that all well-written code shares (see Box 4 for an example):

**Correct**: The code is free from errors, ensuring accurate and expected outcomes.
**Usable**: The code is easy to understand and use.
**Reliable**: The code's output is consistent across different environments.
**Efficient**: The code executes quickly and uses computational resources wisely.
**Adaptable**: The code can easily be applied to problems similar to the one it was originally written for.

It is tempting to rush through coding to quickly move to the core of a project, but taking a deliberate approach to code that focuses on quality and correctness generally pays off in the long run. Investing time in writing clear, high-quality, and extendable code is worthwhile, as it significantly reduces the need for extensive debugging and refactoring in the future. Several timeless principles should be considered during coding to ensure that these criteria are met.

**Keep it short and simple (KISS).** This is the code version of Occam's razor: the simplest version is probably the best. Avoid using excessive abstractions or complex design patterns, especially if the problem at hand does not require them. For example, at the outset of a research project, it is preferable to write small, simple functions that do one thing well, rather than building a complex object-oriented system with multiple layers of inheritance.

**Do not repeat yourself (DRY).** Repeating code should be minimized to avoid complications down the line. Duplicated code is harder to maintain, as changes must be applied across multiple locations, and bugs can propagate throughout the codebase. Duplicates clutter up code, reducing readability and reusability. The easy solution is to encapsulate code into functions or classes and call them as needed.

**Use descriptive names for variables, functions, and classes.** Good code should be self-explanatory, or as Grady Booch said, "read like well-written prose"[47]. Ambiguous variables named like "foo" or "x" can make code hard to follow, whereas descriptive names like "number_of_trials" clearly convey the meaning of a variable. Use names that are pronounceable and searchable but with meaningful distinctions, and avoid very similar names, such as "data1" and "data2".

Generally, it is advisable to stick to a naming convention or style guide of a programming language. These guides reflect the consensus of software engineers and are widely available online (e.g., Google style guides for different languages, including R and Python, https://google.github.io/styleguide/). For example, functions and methods should typically be verbs (e.g., "runExperiment()" or "run_experiment()"), while variables usually start with a lowercase letter. Class names should be nouns, with the first letter of each internal word capitalized (e.g., class "SignalTrial"). Modern IDEs often assist with auto-completion, making it easier to stick to naming conventions.

**Do not hard-code experiment configuration.** A common cause of crashing experiments or analyses is hard-coded configuration variables or file paths. This means that some parameters required by the program are directly embedded within the code itself, such as assigning the display size of an experiment. By doing so, the code becomes tightly coupled to the specific environment in which it was created and is likely not compatible with other systems. Moreover, it is also easy to overlook these hardcoded values, causing bugs that are hard to detect. Instead, consider passing these parameters to the program as command line arguments or storing them in a configuration file that the program reads at the beginning.

While in most cases we encourage the use of relative file paths, there are rare cases where absolute paths are preferred, for example when the relative location of a script or its associated data is not fixed.

## Principle 5: Test your code
*Scenario: You wrote code for an experiment where participants need to respond to stimuli with a button press. Your script looks fine when testing it on yourself, and confidently, you move on to collect data from a larger sample. However, halfway through the data collection, your script crashes when a participant presses two buttons at the same time, an edge case you did not foresee. You now need to remove this participant from your study and stressfully fix your code before the next experiment starts.*

Testing is a standard practice in professional software development to catch these and other kinds of issues. There are manual versions of testing, e.g., participating in your own experiments or reviewing code line-by-line to catch errors. These forms of testing do not prevent edge cases and offer only limited assurance about the code's reliability to other researchers.

Instead, code should be tested automatically at multiple levels, from individual functions (Box 5) to complete analysis pipelines. Start with focusing on core functionalities needed for analyses and keep iterating between implementing and testing new code to expand the codebase. This approach not only ensures that code works as intended, but also minimizes the introduction of new bugs, as any changes that break the code can be identified quickly through failed tests.

We recommend checking out a general introduction to testing and hands-on tutorials about writing tests (like the "Arrange, Act, and Assert" pattern)[48], testing frameworks including *pytest*[49] for Python and the *assertthat*[50] package for R. Researchers more experienced with programming could consider making testing the foundation of their coding ("test-driven development") and fully automating it to run whenever changes are being made to the code ("continuous development/integration")[51].

Additionally, any script that is likely to run for a long time should incorporate checks of key variables at the beginning of the script. For example, this could include confirming if required files actually exist, output directories are writable, or all input data is in the correct format. Neglecting these checks can lead to scripts failing after lengthy computations, wasting time and effort. By validating these factors at the beginning of a script, potential problems can instead be caught and addressed immediately.

## Principle 6: Think about others
*Scenario: Several months into a longitudinal study on cognitive decline, you struggle to adjust your analysis script, finding yourself lost in thousands of*

## Box 5 | Testing

In the "Arrange, Act, and Assert" pattern, the necessary variables to test a function are set up (arrange), the code being tested is executed (act), and the outcome is compared to the expected results (assert). An example of testing a simple function could look like this:

```python
import pytest

# Function to sort a list after removing
duplicates
def unique_sort(input_list):
    if len(input_list) == 0:
        return "List empty!"

    unique_list = list(set(input_list))
    sorted_list = sorted(unique_list)
    return sorted_list

# Tests using the Arrange, Act, and Assert
pattern
def test_unique_sort():
    # Test 1: Does it correctly sort a list
without duplicates?
    # Arrange
    arr = [4, 2, 9, 1, 5, 6]

    # Act - execute the function on the fake data
    sorted_arr = unique_sort(arr)

    # Assert - throws an error when condition is
not met
    assert sorted_arr == [1, 2, 4, 5, 6, 9],
f"Expected [1, 2, 4, 5, 6, 9], got {sorted_arr}"

    # Test 2: Are duplicates correctly removed?
    arr_with_duplicates = [1, 2, 2, 3, 4, 3, 5]
    sorted_arr = unique_sort(arr_with_duplicates)
    assert sorted_arr == [1, 2, 3, 4, 5],
f"Expected [1, 2, 3, 4, 5], got {sorted_arr}"

    # Test 3: Is an empty list handled correctly?
    assert unique_sort([]) == [], "Expected an
empty list"
```

```R
library(testthat)

# Function to sort a list after removing
duplicates
unique_sort <- function(input_list) {
  if (length(input_list) == 0) {
    return("List empty!")
  }

  unique_list <- unique(input_list)
  sorted_list <- sort(unique_list)
}

# Tests using the Arrange, Act, and Assert
pattern
testthat::test_that("Test 1: Does it
correctly sort a list without duplicates?",
{
  expect_equal(unique_sort(c(4, 2, 9, 1, 5,
6)), c(1, 2, 4, 5, 6, 9))
})

testthat::test_that("Test 2: Are duplicates
correctly removed?", {
  expect_equal(unique_sort(c(1, 2, 2, 3, 4,
3, 5)), c(1, 2, 3, 4, 5))
})

testthat::test_that("Test 3: Is an empty
list handled correctly?", {
  expect_equal(unique_sort(c()), c())
})
```

In this example, the third test would fail, as the function does not behave as expected. The test requires it to return an empty list, given an empty list as input, but it returns the string "List empty!" instead.

*lines of undocumented code in a single script. Despite the challenge, you publish your results, generating a great deal of interest. Other researchers reach out, and you happily share the script via a Dropbox folder. Unfortunately, the feedback you receive is disheartening: no one can get it to run, and many abandon trying to decipher the code soon after.*

As scientists, we share a certain responsibility to make research accessible and reproducible. This means not only publishing journal articles but also the data and code underpinning the analyses. The basic prerequisites for sharing code have already been discussed in previous sections: ensuring high quality by following best practices and style guides, standardizing programming environments by documenting dependencies, and testing code to ensure correctness.

Beyond these steps, we need to ensure that code is easy to understand, both by others and our future selves. Code that is flexible and easy to use will lower the barrier for broader adoption in the field, but code that is overly rigid or difficult to use risks being ignored, no matter how valuable it could be. Adopting a mindset of "defensive" programming - anticipating unintended use, such as moving scripts outside of the project directory, unexpected input formats, or missing dependencies - and safeguarding against these scenarios can also make code more robust and adaptable. This section covers recommendations for improving code reusability.

**Document code.** Documentation is key to clarifying the purpose of a program and how to use it. The audience is not only end users—documentation also serves other developers. Forms of documentation include comments, docstrings, examples, README files, user manuals, and descriptive error messages.

Comments are the most basic form of documentation. They should explain the "why" of the code—its intent—rather than the "what" and how it is implemented, except for complex sections. For instance, "compute_mean(reaction_times) # compute the mean of the reaction times" adds no useful information.

Functions should include a docstring that briefly describes their purpose, parameters, and outputs. Beyond inline comments and docstrings, consider creating supplementary documents like environment setup guides, quick-start manuals, and demo examples. Documentation that is outdated can lead to confusion, making it important to update it regularly.

A somewhat different but potentially very useful form of documentation is error messages. All code can produce errors and crash, but default error messages can be cryptic or sometimes misleading. It can be very helpful to anticipate these user-based errors and provide descriptive error messages, stating exactly what the user did wrong and how they can fix it.

For more details about documentation, see Lee[52] for ten rules for documenting software.

**Refactor code**. Refactoring refers to restructuring of code to improve its readability and maintainability, without changing its functionality. Regularly refactoring, especially for long-term projects, allows researchers to adapt their codebase to changing requirements and to keep it organized.

The most important refactoring step for scientists is centralizing scattered functionality. For instance, if two scripts contain identical analysis code, it should be unified into an 'analysis' module (Do not repeat yourself (DRY)). One advantage of refactoring lies in the opportunity that comes with the extraction and reorganization of code: it allows researchers to review code for best practices, including self-explanatory variable names, modular and reusable code, and functions that only perform a single job.

**Create reusable tools instead of one-off scripts**. For researchers who focus on method development, creating reusable tools can help dramatically with adoption by the community. However, every researcher can benefit from transforming scattered research scripts into a maintainable, user-friendly tool. This does require additional effort but offers many advantages both for individual work and potential sharing with others, as improved usability, understandability, and efficiency save time in the long run.

For usability, it is important to keep a tool's complexity from the user: They do not need to know all classes and functions under the hood, nor should they have to piece them together for desired functionality. Providing a simple, extendable interface that covers common use cases can attract new users, while still leaving room for advanced programmers to customize functionality. It also makes it easier to reuse code in new projects as well.

For understandability, it is important to document the user interface and underlying code. Good coding practices support this task, including telling variable names, comments explaining the code, and functions with helpful docstrings. A detailed description of functions for use by others, known as the application programming interface (API), is crucial if the code is to be used by others and can be automatically generated using tools like *Doxygen*[53] or *Sphinx*[54].

Finally, for advanced users, before distributing code as a toolbox, it is important to not only make sure it works as intended but to also ensure it is fast and memory-efficient - the downstream time cost of inefficient code will multiply with the number of other users, and fast and efficient code will spare them frustration.

**Principle 7: Seek awareness and consensus**

*Scenario: Your research team consists of graduate students from various backgrounds, all working on extending a cognitive model of behavior in monetary tasks. One psychology student writes functional but poorly formatted code, arguing that appearance does not matter. Meanwhile, a computer science student strictly follows a style guide, believing it is crucial for maintainability, even if it slows progress. Without consensus on coding standards, conflicts arise, bugs become more frequent, and integrating the team's work becomes challenging.*

Adopting good programming practices can be challenging, specifically in diverse research groups. To get everyone in the team on board, we suggest creating a living document of shared programming practices and dedicating enough time to get everyone on the same page. This document could take a QA format, including questions like:

How do we structure code for new research projects?
Do we want to follow a specific coding style guide?
How do we ensure the code we write is correct?
Will we implement regular code reviews?
How can we make sure our code is easy to use?
What platform will we use to share code?
To what extent should code be documented?

This document can also be extended to cover data management, computing resource usage, and open science practices. Since every research group has unique needs, we encourage readers to identify other areas where establishing a shared consensus would be beneficial.

**Principle 8: Conduct code reviews**

*Scenario: In a collaborative project about the meta-analysis of acoustics perception data, a junior researcher unknowingly introduces a bug in the preprocessing code, skewing all data. This flaw goes unnoticed, leading the team to base significant research on faulty data. Months later, the error is discovered during peer review, forcing the team to retract their results and rerun all analyses. The most experienced programmer in the team attempts to prevent future issues by reviewing all code from colleagues but becomes overwhelmed by the volume, causing delays, limited feedback, team frustration, and declining code quality.*

Code review, a process where programmers evaluate each other's work to identify bugs and improve quality, is surprisingly uncommon in research settings[5]. However, it is a powerful tool, not only for complex team projects but also for individuals. There are two main challenges in academia that are less present in industry. First, much work in academia is focused on individuals and their projects, where there is no strong incentive for collaborators to also engage in code review early on. Second, work in academia is typically less structured than in industry settings, where tasks are more clearly defined and manageable, making them easier to review. In academia, it can be hard to determine the scope of a review, and the amount of code can be overwhelming, especially if reviews only start midway through a project.

To make code review effective, it is important to establish a review protocol early on, agreeing on the timing and scope of reviews. The approach can vary between labs: one lab might prefer an "ad-hoc" review system, where reviews are requested as uncertainties arise, whereas another group might opt for defining tasks in advance, allowing for a thorough review of smaller, more incremental steps.

Deciding who conducts the reviews is also essential. Avoid overburdening experienced programmers, who are often sought after for their knowledge. Instead, we suggest considering systematic approaches like a buddy system (A reviews B, and B reviews A), a chain system (A reviews B, B reviews C, C reviews A), or a group system (A or B reviews C). Each method fosters collaboration and learning, both for the reviewer and the author of the code. Recognizing the work invested into code reviews is very important; we maintain that significant contributions should warrant co-authorship on publications arising from this work if other conditions for authorship are met (for example, consult the authorship rules of the International Committee of Medical Journal Editors, ICMJE).

While some PIs might view internal code reviews as time-consuming, reviews ultimately lead to better code, fewer errors, and increased group productivity. Further insights into code review practices, challenges, and practical advice can be found in several other works[5,55,56].

**Principle 9: Build a shared knowledge base**

*Scenario: A postdoc researcher who developed a specialized preprocessing pipeline for eye-tracking data leaves your lab for another position and takes with them the specific settings, scripts, and reasoning for them. The remaining team members struggle to reconstruct the pipeline, causing delays in follow-up projects and accidentally introducing inconsistencies in data processing methodologies.*

It is common for researchers in cognitive neuroscience and psychology to leave a research group, and that specialized knowledge disappears from the lab. In the biomedical sciences, it is standard practice to document detailed protocols that explain step-by-step how specific processes work, like cell culture maintenance or tissue sample preparation. Specialized knowledge, including but not limited to code, can quickly fade if not preserved, and research groups are no exception. To prevent losing valuable knowledge, it is important to store it in an accessible manner to prevent scenarios like those in the example.

Code should generally be stored on online version control platforms. These platforms not only track changes and back up code - they also support collaboration through features like code reviews via pull/merge requests and

## Box 6 | Actionable checklist

**Code organization and efficiency**:

[ ] Are you using a proper IDE?

[ ] Is the project directory structure clear and consistent, and does it follow the lab's standard?

[ ] Have you documented how to recreate your development environment?

[ ] Have you used existing libraries for common tasks instead of custom solutions? If not, did you cross-check with existing libraries that the code works as expected, is stable even for edge cases, and yields the correct solutions?

[ ] Is all relevant code committed to a version control system?

[ ] Have you checked that there are no repetitive tasks left that could be automated?

[ ] Have you checked for performance bottlenecks?

**Code quality and reusability:**

[ ] Are your function and variable names descriptive and consistent?

[ ] Have you removed any duplicate code by creating functions?

[ ] Have you replaced all hard-coded values with configurable parameters?

[ ] Have you written tests for critical functions, even if they are only assert statements?

[ ] Is your code sufficiently documented (comments, docstrings, README)?

[ ] If applicable: Did you generalize code for future use by others and yourself?

**Collaboration and sharing:**

[ ] Does your code adhere to your lab's shared coding standards?

[ ] Did someone review your code?

[ ] Did you incorporate your new knowledge into a shared wiki?

[ ] Is your code correct and understandable, and can it easily be used to reproduce your research findings?

[ ] If applicable: Did you add relevant code to your lab's shared library?

direct discussions via issues and comments. Free public services, like GitHub, GitLab, or BitBucket, are often suitable, but depending on a lab's data protection policies, private hosting may be required, which an IT department should be able to provide.

In addition to project-specific repositories, we suggest centralizing code for frequent tasks within the group. Given that researchers in a group often work on similar topics or share certain analyses (e.g., loading a lab-generated dataset), streamlining these tasks via a shared software library can save significant time otherwise spent reinventing the wheel.

Beyond code, it is also important to protect other pieces of knowledge in a group from being lost, such as the usage of local resources, or the rationale behind a preprocessing pipeline. If this information is not documented together with related code, it should be included in a shared group wiki. The choice of platform (e.g., Confluence, JupyterBook, Notion) is less critical compared to the group's commitment to maintaining it over time. We recommend that PIs factor in the time it takes for their lab members to properly store and document knowledge to ensure it is preserved. Starting this process early ensures that others can test the documentation and leaves room for improving its usability.

### Principle 10: Consider infrastructure

*Scenario: Your lab relies on outdated local workstations for computationally intensive tasks, like neural network training, neuroimaging data processing, and statistical modeling. These machines are not only slow but also have inconsistent software environments. As a result, researchers often wait for code to run or troubleshoot software issues instead of focusing on their scientific work, significantly prolonging the time to publication and noticeably reducing the lab's overall productivity.*

When research requires substantial computational resources - such as CPUs, GPUs, RAM, or disk storage—these should be considered integral components of a lab's infrastructure. While universities and research institutes typically provide high-performance computing clusters (HPCs) and workstations, these resources often come with their own challenges. HPCs can be difficult to use for non-technical lab members and may sometimes involve unpredictable wait times due to shared resources, specifically when so-called job schedulers like Slurm are used. Local workstations, on the other hand, may be slow, limited in capacity, or suffer from unreliable connections to university servers. Commercial cloud computing service providers offer flexible and on-demand solutions but may be difficult to use for beginners and also raise concerns about data privacy. A good

solution for small to intermediate size labs may be interactive servers, but they require maintenance and awareness about individual resource usage.

Frustrations related to these resources can often go unnoticed by PIs, as lab members tend to report results rather than the challenges in achieving them, or may not even know of the existence of more efficient solutions. PIs should evaluate these difficulties and address them to allow their team to work on actual research rather than grappling with technical hurdles.

### Limitations and optimizations

In this paper, we explored the benefits of adhering to best coding practices and their impact on scientists' ability to efficiently produce reliable and reproducible research. We provided recommendations for individual researchers on organizing code for effective automation and on writing high-quality code that is accurate, efficient, and understandable. Additionally, we offered suggestions for PIs to improve the efficiency and research impact of their lab by fostering a shared understanding of these practices and providing the necessary support to maintain them.

These recommendations are meant as a general guide and are not a one-size-fits-all solution. Research projects vary widely—for example, exploratory analysis needs more flexibility, while large-scale data processing requires a structured approach. The experience level of team members also matters as beginners may find it harder to adopt practices like testing or ensuring high-quality code. To address these variations, we recommend a case-by-case approach: start with practices most relevant to your project or team, gradually integrate others over time, and seek feedback to refine and improve.

While this primer focused on best coding practices, it lacks an in-depth introduction to research data management, which is an important topic on its own. In addition to maintaining a reusable codebase, effectively organizing, storing, and sharing research data plays an important role in ensuring the reproducibility of a research project. Due to space constraints and to keep the topic more focused, we did not cover this topic in detail. We encourage readers interested in best practices for research data management to explore dedicated resources on this topic.

### Outlook

Coding plays an increasingly vital role during all phases of research, from conceptualization of a research idea, over data collection and analysis, up to the publication and dissemination of results. Generally, when coding, alternating between rapidly iterating in prototyping mode to get results quickly and carefully revising and reviewing in development mode to ensure

## Box 7 | Recommendations for further reading

**General advice for research programming:**
- Best practices for scientific computing[11].
- Good enough practices for scientific computing[12].
- The Good Research Code Handbook[58].
- The Turing Way: Guide for reproducible research[61].
- The pragmatic programmer: your journey to mastery[62], Ch. 5 and 7.

**Organizing code**

Principle 1: Adopt sensible standards.
*Quick start*
- The project and data organization section from the RDM Workshop[63] offered a simple and clear overview of this topic (https://julia-pfarr.github.io/rdm_workshop/organization).
- The "Structuring your project" section in the Clean code workshop: MATLAB[64]: https://remi-gau.github.io/matlab_clean_code_workshop/.

*Deep dive*
- The "Structuring data" section from the Research Data Management with DataLad course[65]: https://psychoinformatics-de.github.io/rdm-course/02-structuring-data/index.html.
- Python style guide (e.g., Google Python style guide: https://google.github.io/styleguide/pyguide.html).
- R style guide (e.g., tidyverse style guide: https://style.tidyverse.org/).

Principle 2: Track changes.
*Quick start*
- Git and GitLab Tutorial[66]: https://julia-pfarr.github.io/git-gitlab_tutorial/.

*Deep dive*
- Curating research assets: A tutorial on the Git version control system[67].
- Atlassian tutorials of Git and version control systems: https://www.atlassian.com/git/tutorials.

Principle 3: Accelerate the research workflow.
*Quick start*
- Workshop[51] about continuous integration and test-driven development: https://doi.org/10.5281/zenodo.8119398.
- Peikert and Brandmaier[68] suggested a workflow for R-based data analyses.

*Deep dive*
- High-performance Python: Practical performant programming for humans[44].
- Efficient R programming: A practical guide to smarter programming[45].

**Writing reusable research code**

Principle 4: Write "good" code.
*Quick start*
- Workshop about software and scientific software development[9]: https://doi.org/10.5281/zenodo.7924867.
- Workshop about clean code and factorization[10]: https://doi.org/10.5281/zenodo.7972933.

*Deep dive*
- Clean code: a handbook of agile software craftsmanship[47], Ch. 2, 3, and 6.

Principle 5: Test your code.
*Quick start*
- Workshop introducing software testing[48]: https://doi.org/10.5281/zenodo.8069918.
- The "Testing" section from Clean code workshop: MATLAB[64]: https://remi-gau.github.io/matlab_clean_code_workshop/.

*Deep dive*
- Clean code: a handbook of agile software craftsmanship[47], Ch. 9.
- The "Code testing" section from the NOWA school[69]: https://julia-pfarr.gitlab.io/nowaschool/materials/CI_CD/Code_testing.html.

Principle 6: Think about others.
*Quick start*
- Lee[52] offers easy-to-follow suggestions on code documentation.

*Deep dive*
- Refactoring: improving the design of existing code[70]: https://refactoring.com/.
- Clean code: a handbook of agile software craftsmanship[47], Ch. 4.
- Schilder, Murphy, and Skene[71] developed a package to distribute R code by automating the code publishing workflow.

**Programming in a collaborative setting**

Principle 7: Seek awareness and consensus.
- The section of "Team wide practices" from the road map of code review covers several aspects to be considered when building up lab-wise awareness[72]: https://roadmap.sh/best-practices/code-review.
- Inclusivity in coding practice and teaching[73].

Principle 8: Conduct code reviews.
*Quick start*
- Petre and Wilson[5] write about how code review can be implemented in a research group.

*Deep dive*
- The road map of code review[72] provides a comprehensive description of the code review procedure in the industry.

Principle 9: Build a shared knowledge base.
*Quick start*
- The "Collaborating with Git" course[74]: https://vickysteeves.gitlab.io/collaborating-with-git/collaborating-with-git.html.

*Deep dive*
- Github's tutorial about pull requests: https://docs.github.com/en/pull-requests/collaborating-with-pull-requests.

Principle 10: Consider infrastructure.
- The "data storage and sharing" section from the NOWA school[69]: https://julia-pfarr.gitlab.io/nowaschool/materials/RDM/sections/storage_sharing.html.

---

code correctness strikes a good balance between short-term productivity and long-term maintainability.

The benefits of implementing sound coding practices extend beyond individual projects and labs. Importantly, the broader scientific community stands to gain from a commitment to creating well-tested, shareable code and openly sharing both code and research findings. Robust coding practices, along with open science practices, foster transparency, reproducibility, and cumulative scientific progress, as the reliability of methods and results can be more effectively scrutinized and validated by peers[57].

Publishing code is not just a contribution to the scientific community but also can significantly enhance a researcher's professional standing, as it adds credibility to the work, demonstrating transparency and rigor in the

methodology. Additionally, accessible code can lead to greater visibility within the research community, opening doors to potential collaborations and new opportunities that might not have arisen otherwise. Moreover, it is increasingly common for employers or academic institutions to request code samples during the application process, so having publicly available, well-documented code can provide a competitive edge in securing positions.

To gain an understanding of where an individual researcher or their lab currently stands in terms of these practices, we refer the reader to Box 6. For interested readers who would like to learn more about scientific coding practices, we include a collection of key references in Box 7. We finally encourage readers to review their programming habits and adopt the practices that suit their specific research setting and to explore the extensive literature on software development.

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

## Acknowledgements

We thank Stephanie Pieschl, Yanan Sheng, Lea Lempert, Sofia Navarro-Baéz, Franziska Schaefer, Julia-Katharina Pfarr, and Nina Trenz for providing helpful comments on earlier versions of the manuscript. This work was supported by a research group grant by the Max Planck Society awarded to M.N.H., co-funded by the European Union (ERC Starting Grant project COREDIM ERC-StG-2021-101039712 to M.N.H. and ERC Consolidator Grant SEGMENT 101086774 to T.S.A.W.), and the Hessian Ministry of Higher Education, Science, Research, and the Arts (LOEWE Start Professorship to M.N.H. and Research Cluster "The Adaptive Mind" via the Excellence Program of the Hessian Ministry of Higher Education, Science, Research and the Arts). The funders had no role in the decision to publish or preparation of the manuscript. Views and opinions expressed are those of the author(s) only and do not necessarily reflect those of the European Union or the European Research Council. Neither the European Union nor the granting authority can be held responsible for them.

## Author Contributions

J.R.: Conceptualization, Software, Writing—Original Draft. Y.D.: Conceptualization, Visualization, Writing—Original Draft. F.P.M.: Conceptualization, Writing—Review & Editing. P.K.: Conceptualization, Writing—Review & Editing. T.S.A.W.: Conceptualization, Writing—Review & Editing, Supervision, Funding acquisition. M.N.H.: Conceptualization, Writing—Review & Editing, Supervision, Project administration, Funding acquisition.

## Competing interests

The authors declare no competing interests.
