## [Transparent Peer Review file · Communications Psychology]

Ten principles for reliable, efficient, and adaptable coding in psychology and cognitive neuroscience

Corresponding Author: Mr Johannes Roth

Version 0:

Decision Letter:

Dear Mr Roth,

Thank you for your patience during the peer-review process. Your manuscript titled "Recommendations for programming in experimental psychology and cognitive neuroscience" has now been seen by 3 reviewers, and I include their comments at the end of this message.

The reviewers are enthusiastic about your work but also mention areas in which your work requires revisions. We are very interested in the possibility of publishing your Primer in Communications Psychology, but would like to consider your response to these concerns in the form of a revised manuscript before we make a decision on publication.

To aid you with that task, I have included a marked-up version of your manuscript. Additionally, please make sure that:

1) The Primer should not be longer than 8,000 words excluding display items.

2) Formatting of Primer Articles should as closely as possible follow this checklist <https://www.nature.com/documents/natrev-articleformatguide-methodsprimer.pdf>. For the purpose of first-level headings, please submit your revision using the following (unnumbered) first-level headings:

Abstract
Introduction
Methods
Limitations and optimizations
Outlook

Subsections should also not be numbered.

3) A change in title is highly encouraged (for example: "Reliable, efficient, and adaptable coding for cognitive science and psychology").

In sum, we invite you to revise your manuscript taking into account all reviewer and editor comments.

EDITORIAL POLICIES AND FORMATTING

You will find a complete list of formatting requirements following this link: <https://www.nature.com/documents/natrev-articleformatguide-methodsprimer.pdf>

Please use the checklist to prepare your manuscript for resubmission.

* **TRANSPARENT PEER REVIEW:** Communications Psychology uses a transparent peer review system. This means that

we publish the editorial decision letters including Reviewers' comments to the authors and the author rebuttal letters online as a supplementary peer review file. We publish these records for all accepted manuscripts. However, on author request, confidential information and data can be removed from the published reviewer reports and rebuttal letters prior to publication. If your manuscript has been previously reviewed at another journal, those Reviewers' comments would not form part of the published peer review file.

If you have any questions about any of our policies or formatting, please don't hesitate to contact me.

Please use the following link to submit your revised manuscript and a point-by-point response to the referees' comments (which should be in a separate document to any cover letter):

Link Redacted

We hope to receive your revised paper within 12 weeks; please let us know if you aren't able to submit it within this time so that we can discuss how best to proceed. If we don't hear from you, and the revision process takes significantly longer, we may close your file.

We understand that due to the current global situation, the time required for revision may be longer than usual. We would appreciate it if you could keep us informed about an estimated timescale for resubmission, to facilitate our planning. Of course, if you are unable to estimate, we are happy to accommodate necessary extensions nevertheless.

Please do not hesitate to contact me if you have any questions or would like to discuss these revisions further. We look forward to seeing the revised manuscript and thank you for the opportunity to review your work.

Best regards,

Troby Lui, PhD
Associate Editor
Communications Psychology

REVIEWERS' EXPERTISE:

Reviewer #1: science practice
Reviewer #2: coding/toolbox in psychology/neuroscience
Reviewer #3: science practice

REVIEWERS' COMMENTS:

Reviewer #1 (Remarks to the Author):

The manuscript by Roth et al, titled "Recommendations for programming in experimental psychology and cognitive neuroscience" and submitted to the journal Communication Psychology, presents a practical guide to improving coding practices in cognitive neuroscience.

I encourage standardizing and descriptive filenames to go along with the standardized directory structure. One issue with relying on directory names to provide information is that if the files within the directory move, so does the detailed, descriptive naming. For example, if we had `project_name/data/raw_data/sub01/survey.csv`. "survey.csv" is a bad file name. One would not know where it originated if it ever moved from the folder. "project_name_sub01_servery_raw.csv" would be a much safer name. If that file ever gets moved, you know exactly what it is. This should be touched upon.

Related to this and hardcoding variables, I disagree with using relative paths. If the script moves, the relative path might not be correct, and the script may not work. Or if the script is in the path but the current working directory is not in the script's directory, the script would return an error, which may confuse those with less experience.

More broadly, the paper assumes that nothing (e.g., file locations or directory structures) will be moved. This is a dangerous assumption, especially if data/scripts/etc. are stored or shared between two projects. Also, relative paths should be defined—some may not know that.

It might be useful to show an example of a function that does more than one thing and how to divide it into multiple functions. The authors could also give recommendations on how I/O should be handled.

The authors should emphasize when error checking happens in the code. They should be checked as soon as possible. For example, many software do not check whether the output file path exists or the directory is writable. This leads to all the processing and the program failing at the write. For some processing, especially neuroimaging data, this results in a long time to discover the error and wasted time. This error (unwritable files) should be checked at the program's start before any processing is run.

The authors should also emphasize writing clear error messages. Poorly written error messages can be as frustrating and misleading as no error messages.

In general, a nod to defensive programming could be helpful. It is easy to write code that works for you. But when writing code that others will use, it can be useful to assume someone will use it wrong.

Reviewer #2 (Remarks to the Author):

Summary: The manuscript titled "Recommendations for Programming in Experimental Psychology and Cognitive Neuroscience" aims to provide guidance for researchers, including lab managers, students, and postdoctoral fellows, on best practices in programming within the context of experimental psychology and cognitive neuroscience. The authors present practical recommendations intended to help avoid common pitfalls associated with source code management.

Strengths: The paper is commendably practical in its approach, offering numerous examples and supplementary boxes that enhance the narrative and facilitate understanding. The recommendations align well with established best practices in the field, and the authors have effectively cited relevant literature. Their recommendations align with my own (goodresearch.dev). Overall, the manuscript is well-structured and presents valuable insights for its intended audience.

Areas for Improvement: While the manuscript is strong in many respects, I believe there are two key areas where the authors could enhance their work:

Specificity of Programming Languages: The authors do not clearly define which programming languages their recommendations pertain to, resulting in a lack of specificity. It would be beneficial for the manuscript to focus on a particular language, such as Python or R, or to explicitly state if both will be covered. By narrowing the scope, the authors could provide more concrete advice, such as recommending specific package management tools (e.g., pip, conda, or poetry for Python). The current approach, which oscillates between references to various languages, may confuse readers and dilute the effectiveness of the recommendations. I suggest that the authors select a primary language or clearly delineate their guidance for multiple languages, supplemented by references for those interested in additional languages.

Recommendations Regarding AI Tools: The section addressing the use of AI tools feels a bit naive and does not reflect the current realities of many research labs. The authors advocate for limited use of these tools; the reality is that students under pressure and time crunches will use the tools. Instead of flat-out discouraging the use of AI tools, I recommend that the authors provide an account of how to effectively use the tools in ways that maximize usefulness and minimize damage. This could include suggestions for test-driven development when using AI-generated code, setting limits on the amount of code generated at one time, emphasizing the importance of understanding the capabilities and limitations of these tools, and running workshops within a lab to delineate the responsible use of AI tools. Additionally, the authors should consider discussing specific AI tools, such as GitHub Copilot and various language models, and how they can be utilized responsibly in a learning context.

Conclusion: In conclusion, this manuscript presents valuable recommendations that will benefit the research community in experimental psychology and cognitive neuroscience. However, I believe that addressing the points raised above will significantly enhance the clarity and applicability of the recommendations. I commend the authors for their efforts and encourage them to make these minor revisions before publication.

Thank you for considering my review. I look forward to seeing the revised manuscript.

Recommendation: Accept with minor revisions.

Reviewer #3 (Remarks to the Author):

Roth et al. give in their manuscript "Recommendations for programming in experimental psychology and cognitive neuroscience" valuable advice for programming and how to make the code more reproducible. It is an excellent read and will be beneficial for a broad audience of psychology/cog neuro researchers. I have some recommendations that the authors may want to consider when revising their manuscript.

1. There are many resources given already in the paper, but it would be great if each section had at least one in depth additional resource. This will help readers to dig deeper into each topic that for space reasons can only shallowly be addressed. The appendix can be used for this. As it is resources could be more specifically linked to the structure of the manuscript.

I give some examples below but each section should be complemented by resources.

For example additional workflow papers for R:

<https://doi.org/10.1038/s41467-023-44484-5>

<https://doi.org/10.5964/qcmb.3763>

In section 4 it may be beneficial to point out that there are guidance documents on how to write code in different languages for example: <https://style.tidyverse.org/>

Some resources are whole books. This is always a bit of a downturner. I want to dig deeper on this and there even is a

resource on this, oh it is 400 pages tome. Perhaps more specific chapters can be given in the books that address specific problems.

2. I find the part on git confusing. Git is a locally installed version control system (at least in the case of most users). This part could be elaborated a bit or linked to more resources describing the process. As it is now there is no guidance how to connect the local instance to an online repo. There are links to good tutorials later but as it is I doubt that readers will understand how important Git and associated repositories for collaborative coding are.

3. Some people have pointed out that the conda/Anaconda Repo combination may be problematic. See this stackoverflow post:

<https://stackoverflow.com/questions/74762863/are-conda-miniconda-and-anaconda-free-to-use-and-open-source>
The authors may wish to consider whether they want to include Anaconda/conda or just leave the miniforge link

Version 1:

Decision Letter:

Dear Mr Roth,

Thank you for the resubmission of your revised manuscript titled "Best practices for writing reliable, efficient, and adaptable code in psychology and cognitive neuroscience". I am delighted to say that we are happy, in principle, to publish a version that incorporates some final revisions in Communications Psychology.

We therefore invite you to revise your paper one last time to address a list of editorial requests. At the same time we ask that you edit your manuscript to comply with our format requirements and to maximise the accessibility and impact of your work. To provide detailed guidance, I have attached an edited copy of your work.

EDITORIAL REQUESTS:

Please review our specific editorial comments and requests on the attached copy of your manuscript and in the attached "Editorial Requests Table". Please upload the completed table with your manuscript files as a Related Manuscript file.

SUBMISSION INFORMATION:

OPEN ACCESS:

At acceptance, you will be provided with instructions for completing the open access licence agreement on behalf of all authors. This grants us the necessary permissions to publish your paper. Additionally, you will be asked to declare that all required third party permissions have been obtained, and to provide billing information in order to pay the article-processing charge (APC) [you will receive a waiver token].

Link Redacted

Best regards,

Troyby Lui

Troyby Lui, PhD
Associate Editor
Communications Psychology

Response to reviewers

We would like to thank the editor and the reviewers for their positive feedback and their suggestions for improving the manuscript. We highlight all implemented changes below. The original reviewer remarks are highlighted in *italic font*, our response in regular font, and shorter changes to the manuscript in quotation marks. Finally, changes in the manuscript are highlighted in blue.

Reviewer #1

Remark 1

I encourage standardizing and descriptive filenames to go along with the standardized directory structure. One issue with relying on directory names to provide information is that if the files within the directory move, so does the detailed, descriptive naming. For example, if we had `project_name/data/raw_data/sub01/survey.csv`. “survey.csv” is a bad file name. One would not know where it originated if it ever moved from the folder. “project_name_sub01_survey_raw.csv” would be a much safer name. If that file ever gets moved, you know exactly what it is. This should be touched upon.

We thank the reviewer for this suggestion. We have now added an additional paragraph to Section 3.1.1 (“Use a standardized directory structure.”) emphasizing the importance of using clear, descriptive filenames in addition to a standardized directory structure. Specifically, we now highlight that file names should contain enough information to make them interpretable even if they are moved, and we encourage leaning toward verbosity for clarity.

Added text to Section 3.1.1 (“Use a standardized directory structure.”):

“In addition to adopting a standardized directory structure, paying attention to file naming conventions can pay off in the long run. Descriptive and verbose file names can be very useful, especially when files are moved outside of their original directory. For example, if a file named *behavioral.csv* in the directory `project_name/data/raw_data/sub-01` is accidentally misplaced, it will be difficult to impossible to relate it back to the participant or even the project itself. By contrast, a name like `project_name_sub-01_behavioral_raw.csv` conveys the file’s content, subject, and origin.”

Remark 2

Related to this and hardcoding variables, I disagree with using relative paths. If the script moves, the relative path might not be correct, and the script may not work. Or if the script is in the path but the current working directory is not in the script’s directory, the script would return an error, which may confuse those with less experience.

We appreciate the reviewer raising this concern. We have expanded the discussion in Section 4.1.4 (“Do not hard-code experiment configuration.”) to clarify the trade-offs between relative and absolute paths. While relative paths can simplify reproducibility in certain setups, we agree with the reviewer that they can cause problems when files are moved or scripts are executed in different working directories. To address this, we now outline these risks and highlight when using absolute paths might be the more appropriate choice.

Added text to Section 4.1.4 (“Do not hard-code experiment configuration.”):

“However, it can be convenient to use hard-coded file paths when a project’s structure is unlikely to change and files are contained within the project, for example to simplify data loading. In this case, using relative file paths allows moving the whole project to a different environment, keeping relationships between files intact. On the other hand, when the relative location of a script or its associated data is not fixed, absolute paths can be a more reliable choice. Thus, while in most cases we encourage the use of relative file paths, there are also cases where absolute paths are preferred.”

Remark 3

More broadly, the paper assumes that nothing (e.g., file locations or directory structures) will be moved. This is a dangerous assumption, especially if data/scripts/etc. are stored or shared between two projects. Also, relative paths should be defined—some may not know that.

and

In general, a nod to defensive programming could be helpful. It is easy to write code that works for you. But when writing code that others will use, it can be useful to assume someone will use it wrong.

We thank the reviewer for pointing this out. We have added a brief discussion in Section 4.3 (“Principle 6: Think about others.”), advising readers to be mindful about potential issues that could arise when collaborating or sharing resources. We have also defined both relative and absolute paths in Box 2.

Added text to Section 4.3 (“Principle 6: Think about others.”):

“Adopting a mindset of “defensive” programming - anticipating unintended use, such as moving scripts outside of the project directory, unexpected input formats or missing dependencies - and safeguarding against these scenarios can also make code more robust and adaptable.”

Remark 4

It might be useful to show an example of a function that does more than one thing and how to divide it into multiple functions.

We appreciate this suggestion and have updated the code example in Box 4 to also demonstrate how a function performing multiple tasks can be refactored into small, modular functions.

Remark 5

The authors could also give recommendations on how I/O should be handled.

We thank the reviewer for this suggestion and agree that I/O and, more broadly, data management are important aspects of research programming. However, we decided to specifically focus only on code-related advice, as including data management would have significantly expanded the scope and length of this paper. We now clarify this in the introduction.

Added text to Introduction:

“While reproducible data and standards for data management are also important, here we focus on best coding practices but provide references for improving data reproducibility.”

Remark 6

The authors should emphasize when error checking happens in the code. They should be checked as soon as possible. For example, many software do not check whether the output file path exists or the directory is writable. This leads to all the processing and the program failing at the write. For some processing, especially neuroimaging data, this results in a long time to discover the error and wasted time. This error (unwritable files) should be checked at the program's start before any processing is run.

We fully agree with the reviewer on this and have now added this advice to Section 4.2 (“Principle 5: Test your code.”), specifically recommending early checks of important script parameters.

Added text to Section 4.2 (“Principle 5: Test your code.”):

“Additionally, any script that is likely to run for a long time should incorporate checks of key variables at the beginning of the script. For example, this could include confirming if required files actually exist, output directories are writable, or all input data is in the correct format. Neglecting these checks can lead to scripts failing after lengthy computations, wasting time and effort. By validating these factors at the beginning of a script, potential problems can instead be caught and addressed immediately.”

Remark 7

The authors should also emphasize writing clear error messages. Poorly written error messages can be as frustrating and misleading as no error messages.

We appreciate this feedback and added a discussion of error messages as an additional form of documentation to Section 4.3.1 (“Document code.”).

Added text to Section 4.3.1 (“Document code.”):

“A somewhat different, but potentially very useful form of documentation are error messages. All code can produce errors and crash, often without fault of the developer (e.g. wrong variable type for a function's parameter), but default error messages can be cryptic or sometimes misleading. It can be very helpful to anticipate these user-based errors and provide descriptive error messages, stating exactly what the user did wrong and how they can fix it.”

Reviewer #2

Remark 1

The authors do not clearly define which programming languages their recommendations pertain to, resulting in a lack of specificity. It would be beneficial for the manuscript to focus on a particular language, such as Python or R, or to explicitly state if both will be covered. By narrowing the scope, the authors could provide more concrete advice, such as recommending specific package management tools (e.g., pip, conda, or poetry for Python). The current approach, which oscillates between references to various languages, may confuse readers and dilute the effectiveness of the recommendations. I suggest that the

authors select a primary language or clearly delineate their guidance for multiple languages, supplemented by references for those interested in additional languages.

We thank the reviewer for raising this important point. To address this, we now define Python and R as the primary programming languages in the introduction and provide examples and recommendations for both languages side-by-side.

Remark 2

Recommendations Regarding AI Tools: The section addressing the use of AI tools feels a bit naive and does not reflect the current realities of many research labs. The authors advocate for limited use of these tools; the reality is that students under pressure and time crunches will use the tools. Instead of flat-out discouraging the use of AI tools, I recommend that the authors provide an account of how to effectively use the tools in ways that maximize usefulness and minimize damage. This could include suggestions for test-driven development when using AI-generated code, setting limits on the amount of code generated at one time, emphasizing the importance of understanding the capabilities and limitations of these tools, and running workshops within a lab to delineate the responsible use of AI tools. Additionally, the authors should consider discussing specific AI tools, such as GitHub Copilot and various language models, and how they can be utilized responsibly in a learning context.

We appreciate this feedback and agree that our previous recommendations did not reflect the reality of AI use. While we maintain our view that AI-generated code should be carefully checked, we now revised Box 3 to differentiate advice between beginners and more experienced programmers and added examples for different use cases of these tools. We also included a nod to test-driven development as an effective way of ensuring correctness.

Reviewer #3

Remark 1

There are many resources given already in the paper, but it would be great if each section had at least one in depth additional resource. This will help readers to dig deeper into each topic that for space reasons can only shallowly be addressed. The appendix can be used for this. As it is resources could be more specifically linked to the structure of the manuscript.

- *I give some examples below but each section should be complemented by resources. For example additional workflow papers for R:
<https://doi.org/10.1038/s41467-023-44484-5>
<https://doi.org/10.5964/qcmb.3763>*
- *In section 4 it may be beneficial to point out that there are guidance documents on how to write code in different languages for example: <https://style.tidyverse.org>*
- *Some resources are whole books. This is always a bit of a downturner. I want to dig deeper on this and there even is a resource on this, oh it is 400 pages tome. Perhaps more specific chapters can be given in the books that address specific problems.*

We thank the reviewer for these suggestions. To address these points, we ensured that each of our proposed principles has at least one accessible, introductory resource and one in-depth reference (e.g. books), where we now also specifically highlight individual chapters.

We also added a reference to the Google style guides in Section 4.1.3 (“Use descriptive names for variables, functions and classes.”).

Added text to Section 4.1.3 (“Use descriptive names for variables, functions and classes.”):
“These guides reflect the consensus of software engineers and are widely available online (e.g. the Google style guides for different languages, including R and Python: <https://google.github.io/styleguide/>).”

Remark 2

I find the part on git confusing. Git is a locally installed version control system (at least in the case of most users). This part could be elaborated a bit or linked to more resources describing the process. As it is now there is no guidance how to connect the local instance to an online repo. There are links to good tutorials later but as it is I doubt that readers will understand how important Git and associated repositories for collaborative coding are.

We thank the reviewer for highlighting this. We revised Section 3.2 (“Principle 2: Track changes.”) to give a clearer introduction to Git and explicitly explain the connection between local and remote Git repositories, emphasizing their importance for collaborative programming.

Remark 3

Some people have pointed out that the conda/Anaconda Repo combination may be problematic. See this stackoverflow post: <https://stackoverflow.com/questions/74762863/are-conda-miniconda-and-anaconda-free-to-use-and-open-source> The authors may wish to consider whether they want to include Anaconda/conda or just leave the miniforge link

We agree with the reviewer that recommending Anaconda (and the default conda channel) is problematic. In Section 3.1.3 (Prefer existing tools and do not reinvent the wheel.), we now suggest conda-forge instead and briefly explain the licensing issues around Anaconda. We now also reference mini-forge as an alternative installer for conda in Section 3.1.2 (“Configure and save the programming environment.”).

Added text to Section 3.1.3 (“Prefer existing tools and do not reinvent the wheel.”):
“Python users can find a wide array of packages on PyPI (Python Package Index; <https://pypi.org/>) and through volunteer-maintained conda channels, such as conda-forge (<https://conda-forge.org>), both of which are repositories used by package managers (see section 3.1.2 for more details about package management). Conda channels maintained by volunteer communities are always free to use by anyone, whereas the default Conda channel (managed by Anaconda, a for-profit organization) has required payment from certain organizations since 2020 (Clements, 2023).”